# Pt-Amorphous Barium Aluminum Oxide/Carbon Catalysts for an Enhanced Methanol Electrooxidation Reaction

**Tzu Hsuan Chiang \*** , **Wan-Yu Hou, Jia-Wei Hsu and Yu-Si Chen**

Department of Energy Engineering, National United University, No. 2, Lienda, Nan-Shi Li, Miaoli 36006, Taiwan;
a0978880322@gmail.com (W.-Y.H.); adamant0989858384@gmail.com (J.-W.H.); chihippig@gmail.com (Y.-S.C.)
**\*** Correspondence: thchiang@nuu.edu.tw; Tel.: +886-373-82385

**Abstract:** A new type of amorphous barium aluminum oxide was synthesized using a polyol thermal method involving a mixture with Vulcan XC-72 carbon and supported with 20%Pt catalysts to enhance the activity of a methanol electrooxidation reaction (MOR). The maximum current density, electrochemically active surface area (ECSA), and electrochemical impedance spectra (EIS) of the obtained catalysts for MOR were determined. The MORs of barium aluminum oxide with different calcination temperatures and Ba and Al contact ratios were studied. The MOR of the uncalcined amorphous $Ba_{0.5}AlO_x$ catalysts prepared with a mole ratio of 2/1 Ba/Al mixed with Vulcan XC-72 carbon and supported with 20%Pt catalyst (Pt-$Ba_{0.5}AlO_x$/C) was enhanced compared with that of 20%Pt-$Al_2O_3$/C and 20%Pt/C catalysts due to its obtained largest maximum current density of 3.89 mA/cm$^2$ and the largest ECSA of 49.83 m$^2$/g. Therefore, Pt-$Ba_{0.5}AlO_x$/C could provide a new pathway to achieve a sufficient electrical conductivity, and possible synergistic effects with other active components improved the catalytic activity and stability of the prepared catalyst in MOR.

**Keywords:** amorphous; synergistic; methanol electrooxidation

---

## 1. Introduction

In anode, electrocatalysts of direct methanol fuel cells (DMFCs), platinum (Pt) [1], and Pt alloys, such as PtRu [2,3], PtRuCuW [4], PtPd [5], $Pt_{45}Ru_{45}M_{10}$/C (M = Fe, Co, and Ni) [6], have been used for a methanol electrooxidation reaction (MOR). However, these precious metals are poisoned by carbon monoxide (CO) during MOR, causing a rapid decrease in catalytic activities and producing a low power density of DMFCs. Therefore, anode electrodes need a high loading amount of Pt or Pt alloys to maintain the DMFC performance, but this requirement noticeably increases the cost of a whole fuel cell system. A commercial 20%Pt/C catalyst is most commonly used as cathode catalysts of polymer electrolyte fuel cell (PEMFC), DMFC, and other fuel cells [7]. Advanced supports with synergistic effects for Pt catalysts should be developed by loading Pt on metal oxide catalysts/C catalysts to improve the CO tolerance, reduce Pt loading, and increase electrochemical durability; some metal oxides that have been studied are $CeO_2$ [8], $SnO_x$ [9], $TiO_2$, $Al_2O_3$ [10], and $Nb_2O_5$ [11] because they have a facile oxygen species-releasing ability and good corrosion resistance [11]. In addition, the combination of two or more transition metal oxides, which are better than simple individual oxides, can synergistically enhance the catalytic activity of methanol oxidation [11]. Some bimetallic oxides, such as $NiMoO_4$ [12], Ni-doped $CeO_2$ [13], $Ce_{0.2}Mo_{0.8}O_{3-\delta}$ [14], $SrMO_3$ (M = Ti, Ru) [15], and $Ce_xZr_{1-x}O_2$ [16], have also been explored.

Among various oxides, $Al_2O_3$ has been able to promote a high proportion of metallic Pt species and oxidation state of Pt formation, resulting in Pt/$Al_2O_3$ that has good high activity for CO oxidation [17]

and enhanced MOR activity. In addition, amorphous $Al_2O_3$ has a significant density of negative charges [18], which are easily adsorbed on carbon black with positive surface charges [19] to form a strongly hydrophilic hybrid support and improve Pt deposition, which should play an important role in determining the catalytic activity. $Al_2O_3$ is modified by doping with other metals, such as Mo [20] and Ru [21]. This approach is effective in refining the physical and chemical properties of $Al_2O_3$. This study proposed the use of bimetallic oxide with barium (Ba) precursor as a dopant to enhance the catalytic activity of $Al_2O_3$ based on synergistic effects. This proposal was also based on barium oxide (BaO)-modified Pd-based catalyst that promoted methanol conversion [22]. BaO is also a promoter of $La_{1.6}Ba_{0.4}NiO_4$ catalysts that improve the catalytic activity of NO direct decomposition [23]. Amorphous barium aluminum oxide catalysts were successfully prepared via a facile polyol thermal method and combined with carbon black to support 20 wt.% of Pt particles as an electrocatalyst (20%Pt-BaAlO$_x$/C) with improved activity, stability for MOR, and $CO_{ads}$ tolerance. Different calcining temperatures and different contact ratios of Ba and Al precursors were also discussed in detail.

## 2. Results and Discussion

### 2.1. Effect of Different Calcining Temperatures on the MOR of the Catalyst

Figure 1a illustrates the cyclic voltammetry (CV) of Pt-Ba$_{0.5}$AlO$_x$/C catalysts with Ba$_{0.5}$AlO$_x$ exposed to different calcining temperatures. All the catalysts had similar forward peak potentials, namely, 0.66, 0.65, and 0.65 V. However, the maximum current density in the forward sweep of CVs for $CH_3OH$ oxidation and electrochemically active surface area (ECSA) are important properties for determining elements in the evaluation of the MOR activity of an electrode in DMFCs. A catalyst with a large ECSA usually possesses a high electrocatalytic activity [24]. In Figure 1a,b, the performance of the obtained uncalcined Ba$_{0.5}$AlO$_x$ catalyst was higher than that of the two other catalysts at calcining temperatures of 200 °C and 400 °C. This result indicated that the maximum current density (3.89 mA/cm$^2$) and ECSA (49.83 m$^2$/g) for MOR of the uncalcined catalysts were higher than those of the catalysts calcined at 200 °C (2.89 mA/cm$^2$ and 25.12 m$^2$/g) and 400 °C (1.08 mA/cm$^2$ and 22.52 m$^2$/g). The charge transport properties of various catalysts were assessed by using the Nyquist plots of the electrochemical impedance spectra (EIS) measurement; the diameter of the primary semicircle is closely related to charge reaction resistances (R$_{ct}$) associated with MOR [25]. The small diameter of the semicircle corresponds to the high electrocatalytic activity of a catalyst for MOR. As shown in Figure 1c, the diameter of the primary semicircle for the impedance data could be modeled in the equivalent circuit model illustrated in Figure 1d [25,26]. The circuit elements used were as follows: R$_s$ is the solution resistance, R$_o$ is the contact resistance between the catalyst and the support electrode, and a constant phase element (CPE) is the double-layer capacitance, which is associated with the adsorption of intermediates formed during MOR [25]. In Figure 1c, R$_{ct}$ of 20%Pt-Ba$_{0.5}$AlO$_x$/C catalysts prepared with the uncalcined Ba$_{0.5}$AlO$_x$ was smaller than that of the catalysts calcined at 200 °C and 400 °C. These results indicated that the ion transfer rate of the MOR of the 20%Pt-Ba$_{0.5}$AlO$_x$/C catalyst prepared by uncalcining Ba$_{0.5}$AlO$_x$ was faster than that of the two other catalysts calcined at 200 °C and 400 °C because of the presence of conductive carbon support [12]. Thus, the best electrocatalytic activity of the catalyst for MOR was obtained.

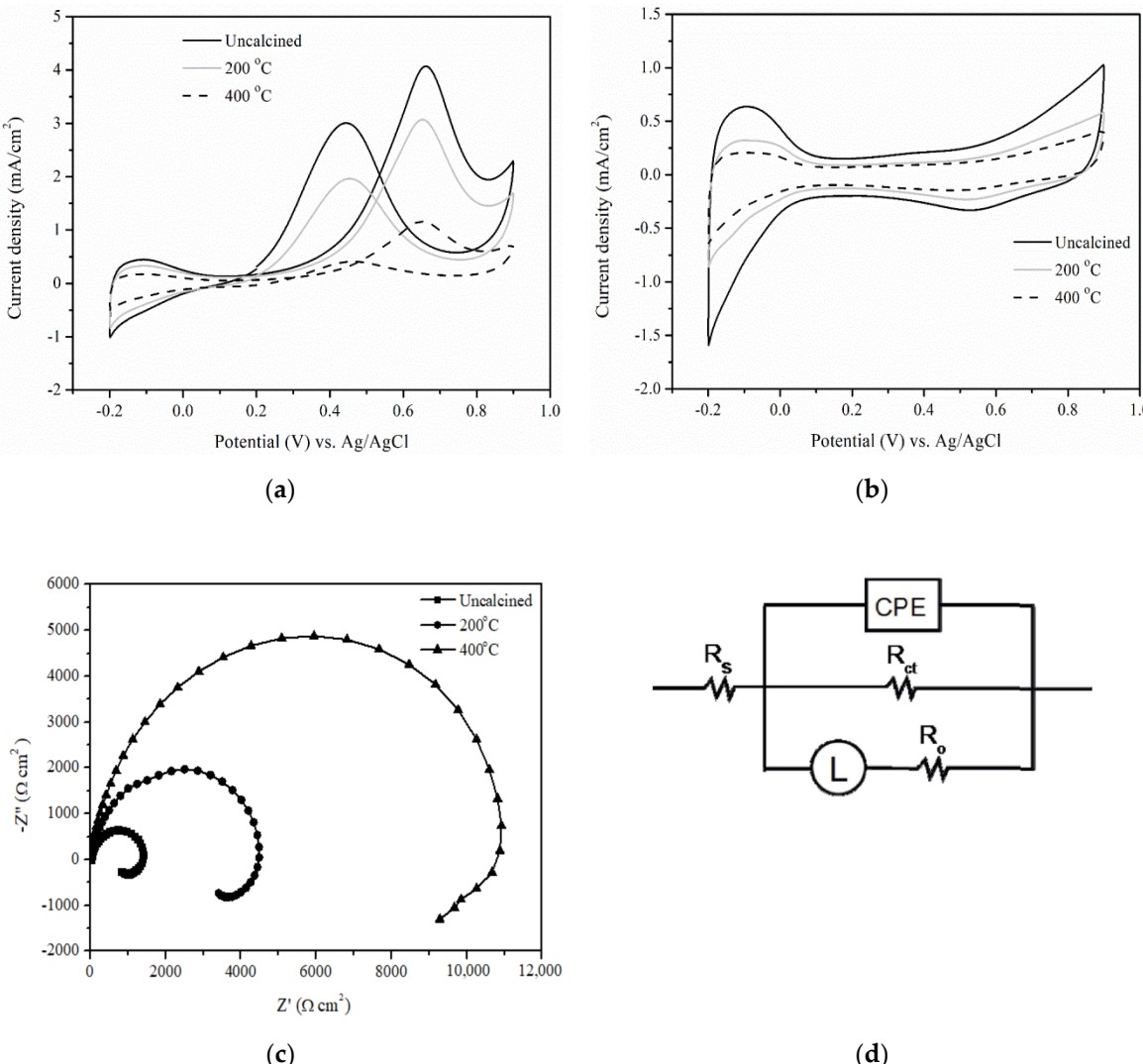

**Figure 1.** (**a**) CV curves in 1 M $CH_3OH$ + 0.5 M $H_2SO_4$, (**b**) in 0.5 M $H_2SO_4$, (**c**) EIS (electrochemical impedance spectra) of 20%Pt-$Ba_{0.5}AlO_x$/C prepared with $Ba_{0.5}AlO_x$ calcined at different temperatures, and (**d**) electrical equivalent circuit model.

Amorphous structures were generated when uncalcined $Ba_{0.5}AlO_x$ catalysts and catalysts calcined at 200 °C to 400 °C were used, as revealed by the XRD data and TEM image shown in Figure 2a,b. However, their particle size distributions differed, suggesting that particle size affected catalyst activities. The histograms of the particle size distribution of the uncalcined $Ba_{0.5}AlO_x$ catalysts and the catalysts calcined at 200 °C to 400 °C are shown in Figure 3. The results indicated that the maximum average particle size ranged from 0.6 µm to 3.0 µm, 0.6, 2.0, and 8.0 µm, 5.0 µm to 10.5 µm, respectively. Their specific surface areas were 7.79 ± 0.01, 5.24 ± 0.01, and 3.48 ± 0.01 $m^2$/g, which corresponded to the small particle sizes of catalysts with large specific surface areas. These results elucidated the effect of the observed particle size on MOR. Bergamaski et al. [27] reported the increasing Pt-CO and Pt-OH bond strength with decreasing particle size on MOR. Therefore, the effect of the particle size of $Ba_{0.5}AlO_x$ observed on the MOR was due to the increased bond strength of $Ba_{0.5}AlO_x$-$OH_{ads}$ with a decrease in their particle size, leading to a higher MOR. Therefore, the MOR activity decreased with the calcined temperature of $Ba_{0.5}AlO_x$ catalysts, with an increasing influence on increased particle size and decreased surface areas. The results obtained showed that the uncalcined $Ba_{0.5}AlO_x$ catalysts possessed the outstanding MOR activity, having the smallest particle size and the largest specific surface areas.

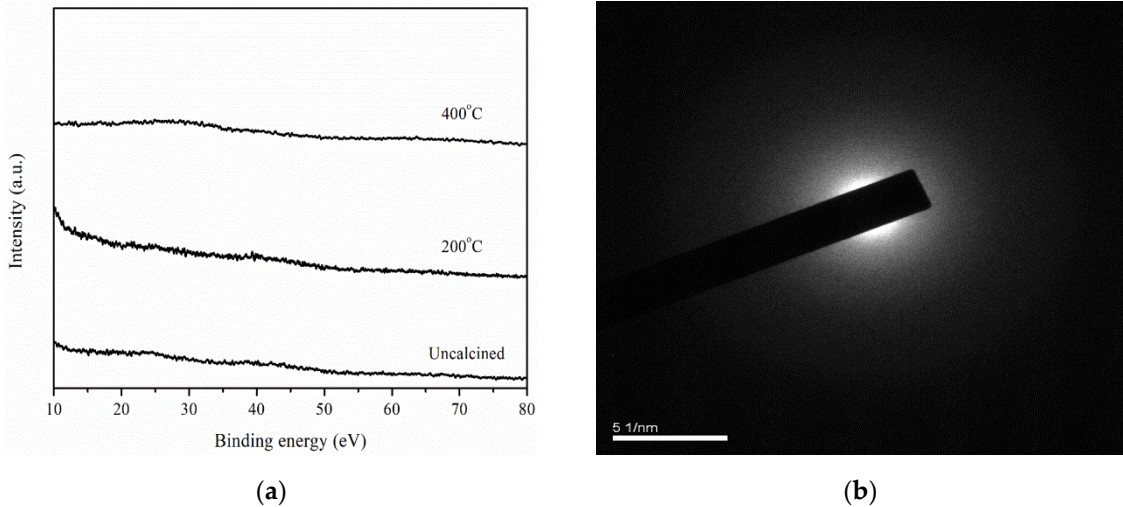

(**a**)　　　　　　　　　　　　　　　　　　　　　　　　(**b**)

**Figure 2.** (**a**) XRD data of $Ba_{0.5}AlO_x$ under different calcining temperatures and (**b**) TEM image of uncalcined $Ba_{0.5}AlO_x$.

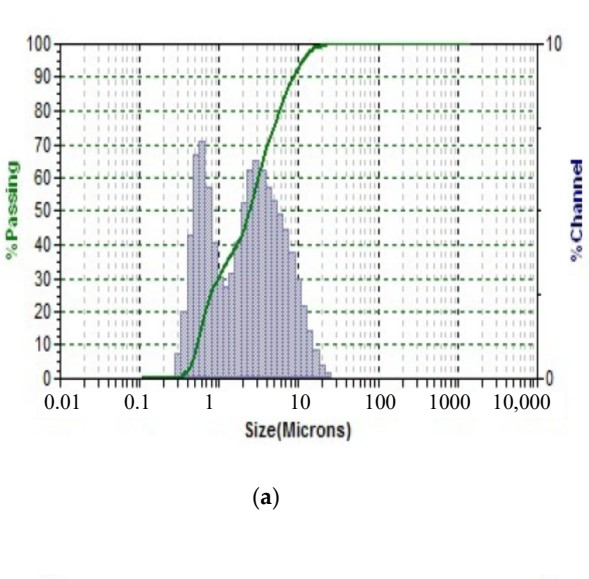

(**a**)

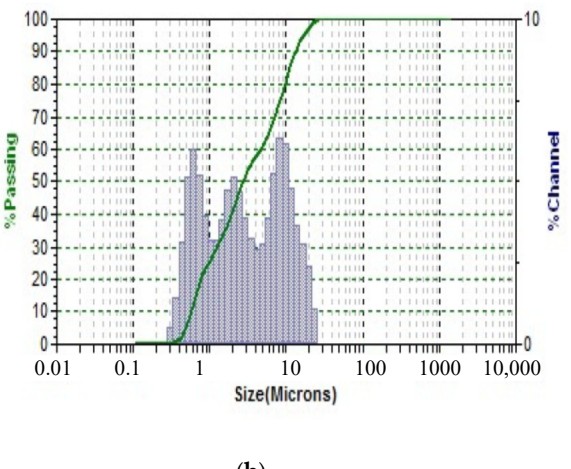

(**b**)

**Figure 3.** *Cont.*

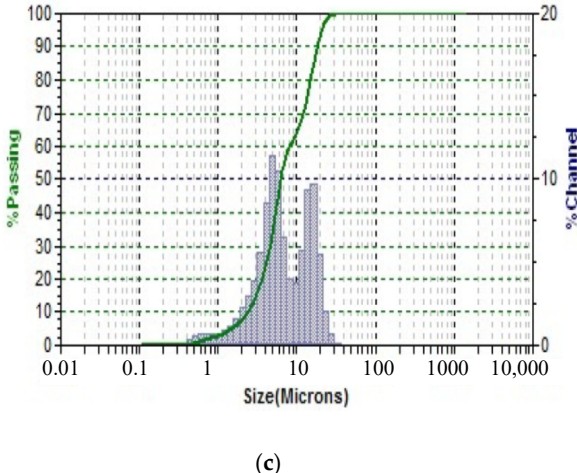

(c)

**Figure 3.** The particle size distribution of (**a**) uncalcined $Ba_{0.5}AlO_x$, and $Ba_{0.5}AlO_x$ powders calcined at (**b**) 200 °C and (**c**) 400 °C.

O 1s, Ba 3d, and Al 2p XPS of $Ba_{0.5}AlO_x$ under different calcining temperatures are shown in Figure 4. The binding energies of O 1s and Ba 3d were similar in the uncalcined $Ba_{0.5}AlO_x$ catalysts or the catalysts calcined at 200 °C and 400 °C. The peak at a binding energy of O 1s for each catalyst displayed the aluminum hydroxide $(Al(OH)_3)$ or aluminum oxyhydroxide state $(AlO(OH))$ at 532.4 eV and aluminum oxide $(Al_2O_3)$ at 531.33 eV [28]. The existence of −OH possibly causes $Al_2O_3$ to react with water, forming $Al(OH)_3$ [29]. In this reaction, water is from the catalyst exposed to the atmosphere (moisture) before XPS measurement [30] and $Al_2O_3$ that reacts with $Al(OH)_3$ to generate $AlO(OH)$ [29]. The peaks at binding energies of 780.4 and 795.5 eV for all the catalysts can be assigned to Ba $3d_{5/2}$ and $3d_{3/2}$ core lines of $Ba^{2+}$, respectively [31]. Particularly, the binding energies of Al 2p XPS varied among the $Ba_{0.5}AlO_x$ catalysts calcined at different temperatures. Only one peak binding energy at 74.1 eV was obtained for the uncalcined $Ba_{0.5}AlO_x$. However, when the calcining temperature increased to 200 °C and 400 °C, the peaks of the binding energies of Al 2p positively shifted to 75.3 and 75.8 eV, which were assigned to $Al(OH)_3$ and $AlO(OH)$, respectively [31].

## 2.2. Effect of Different Ratios of Ba and Al Precursors on MOR

The CVs of MOR at −0.2 V to 0.9 V on 20%Pt with various $BaAlO_x/C$ catalysts composed of $BaAlO_x$ were prepared with different ratios of Ba and Al precursors: 1/1, 1/2, 1/3, 2/1, and 3/1 mole ratios named $Ba_{0.5}Al_{0.5}O_x$, $Ba_{0.5}AlO_x$, $Ba_{0.33}AlO_x$, $BaAl_{0.5}O_x$, and $BaAl_{0.33}O_x$, respectively, as shown in Figure 5a. The forward peak potentials of $BaAlO_x$ catalysts prepared with different ratios of Ba and Al contact on 20%Pt-various $BaAlO_x/C$ catalysts observed at around 0.66–0.67 V in the forward scan (Table 1) were characteristic of MOR through the oxidation of $Pt-(CH_3OH)_{ads}$ on a catalyst surface [32]. However, the forward peak potentials of 20%Pt-$Al_2O_3$/C and 20%Pt/C negatively shifted to 0.64 and 0.65 V compared with that of 20%Pt-$Ba_{0.5}AlO_x$/C catalysts, which indicated that their MOR activity was low. By contrast, the backed peak potentials of various catalysts at around 0.46–0.48 V were similar to that of 20%Pt/C (0.47 V) in the reverse scan and primarily associated with the removal of the residual carbon species formed in the forward scan [33]. Simultaneously, these $BaAlO_x$ catalysts could improve the Pt surface and would be covered with methanol, causing increased H chemisorption on bare Pt, such as $PtOH_{ad} + H^+ + e^- \rightarrow Pt + H_2O$ [34]. Furthermore, Table 1, obtained from Figure 5a, shows that the maximum current densities at the forward and backward peak potentials of 20%Pt-$Al_2O_3$/C catalysts were 0.58 and 0.27 mA/cm², which were lower than those of 20%Pt/C catalysts (1.47 and 0.4 mA/cm²). This result indicated that $Al_2O_3$ was not catalytically active in MOR. Therefore, it obtained the smallest ECSA and the largest EIS, as shown in Figure 5b,c, respectively. However, the maximum current density at the forward and backward peak potentials of $BaAlO_x$ catalysts prepared with different ratios of Ba mixed with Al was higher than that of $Al_2O_3$. These results suggested that

the presence of Ba in BaAlO$_x$ catalysts facilitated the oxidation of freshly chemisorbed species and led to an enhanced MOR through which BaAlO$_x$ catalysts provided active oxygen for the effective removal of intermediates, such as CO on the Pt surface. The maximum current density (4.07 mA/cm$^2$) at the forward peak of 20%Pt-Ba$_{0.5}$AlO$_x$/C (Ba/Al mole ratio of 1/2) catalysts was higher than those of 20%Pt/C and other BaAlO$_x$ catalysts prepared with various mole ratios of Ba and Al contact. These results showed that methanol was more easily oxidized with 20%Pt-Ba$_{0.5}$AlO$_x$/C catalyst than with other catalysts, including the 20%Pt/C catalyst.

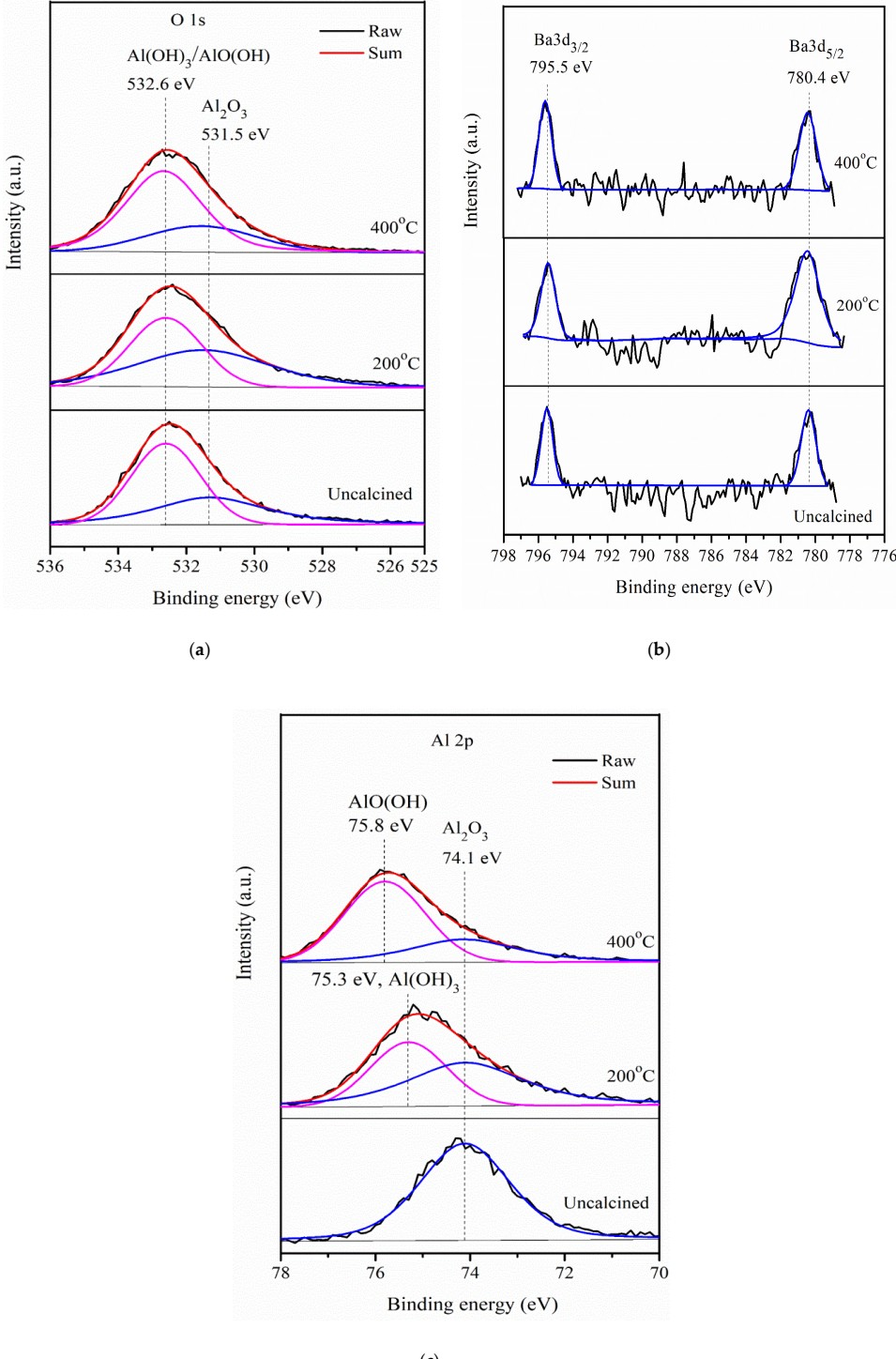

**Figure 4.** (**a**) O 1s (**b**) Ba 3d, and (**c**) Al 2p XPS of Ba$_{0.5}$AlO$_x$ under different calcining temperatures.

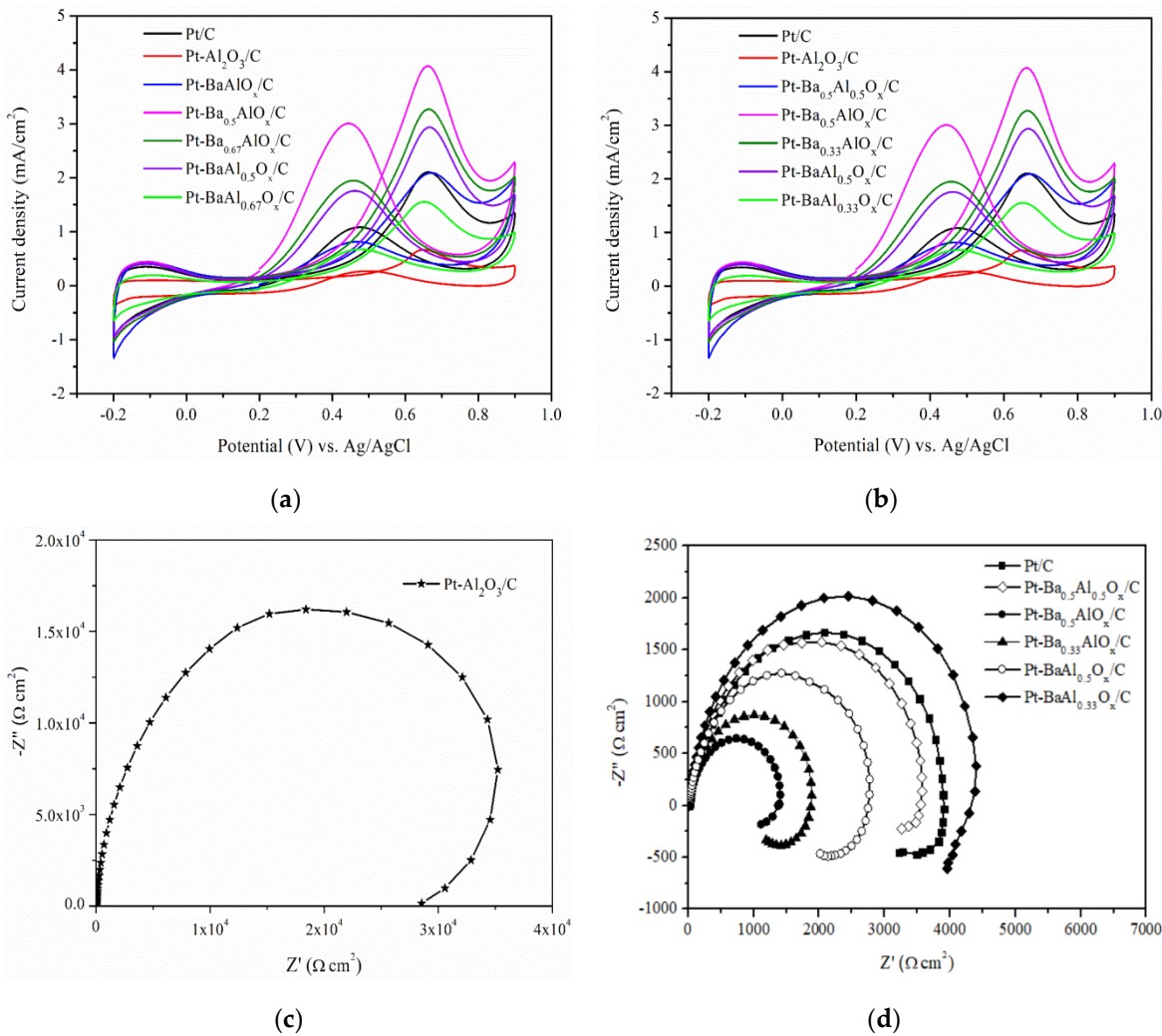

**Figure 5.** (**a**) CV curves in 1 M CH₃OH + 0.5 M H₂SO₄ and (**b**) in 0.5 M H₂SO₄. Nyquist plots of (**c**) 20%Pt-Al₂O₃/C catalysts and (**d**) various catalysts.

**Table 1.** Maximum current density, potential, and electrochemically active surface area (ECSA) of different catalysts in methanol electrooxidation reaction (MOR).

| Catalysts | Maximum Current Density (mA/cm²)/Potential (V) | | ECSA (m²/g) |
|---|---|---|---|
| | Forward Peak (I$_f$) | Backward Peak (I$_b$) | |
| 20%Pt/C | 1.47/0.65 | 0.4/0.47 | 33.02 |
| 20%Pt-Al₂O₃/C | 0.58/0.64 | 0.27/0.51 | 4.78 |
| 20%Pt-Ba$_{0.5}$Al$_{0.5}$O$_x$/C | 1.79/0.67 | 0.42/0.48 | 26.46 |
| 20%Pt-Ba$_{0.5}$AlO$_x$/C | 4.07/0.66 | 2.4/0.46 | 49.83 |
| 20%Pt-Ba$_{0.33}$AlO$_x$/C | 3.07/0.66 | 1.42/0.46 | 42.02 |
| 20%Pt-BaAl$_{0.5}$O$_x$/C | 2.71/0.66 | 1.32/0.46 | 33.08 |
| 20%Pt-BaAl$_{0.33}$O$_x$/C | 1.87/0.67 | 0.77/0.48 | 28.58 |

Furthermore, the ECSA of the different catalysts is shown in Figure 5b and Table 1. The results showed that the MOR of 20%Pt-Ba$_{0.5}$AlO$_x$/C catalysts was dependent on the largest ECSA (49.83 m²/g) among the catalysts. In Figure 5d, R$_{ct}$ of 20%Pt-Ba$_{0.5}$AlO$_x$/C catalysts was lower than that of 20%Pt/C catalysts and other catalysts, and this parameter corresponded to the high electrocatalytic activity of MOR.

Figure 6 shows the XRD data of various $BaAlO_x$ catalysts with different ratios of Ba and Al contact. Amorphous structures formed when the mole ratio of Al precursor contacts was greater than or equal to that of Ba precursors, such as $Ba_{0.5}Al_{0.5}O_x$ (Ba/Al = 1/1), $Ba_{0.5}AlO_x$ (Ba/Al = 1/2), and $Ba_{0.33}AlO_x$ (Ba/Al = 1/3). A crystallization compound that consisted of crystalline $Al_2O_3$ (CCDS11-0517) and $Ba_2Al_2O_5$ (44-0474) structures generated by the mole ratio of Ba precursors was greater than that of the mole ratio of Al precursors, such as $BaAl_{0.5}O_x$ (Ba/Al = 2/1) and $BaAl_{0.33}O_x$ (Ba/Al = 3/1). Unfortunately, the MOR activity of these crystalline $BaAlO_x$ catalysts was lower than amorphous $BaAlO_x$ catalysts because amorphous $BaAlO_x$ catalysts were possibly composed of amorphous $Al_2O_3$ existing in neutral oxygen vacancies [35]. Abundant oxygen vacancies exhibit a dramatically improved electrocatalytic activity of MOR [36]. As more oxygen vacancies are generated, they lead to enhance the surface oxygen concentration on metal oxide catalysts, providing more metal oxide-$OH_{ads}$ bond [37], which boosts the conversion of $CO_{ads}$ to $CO_2$ on the Pt-metal oxide/C catalyst [38]. Therefore, the oxygen vacancies of the $Ba_{0.5}AlO_x$ catalyst were more than those of the $Ba_{0.5}Al_{0.5}O_x$ catalyst because the Al mole ratio of $Ba_{0.5}AlO_x$ was higher than that of $Ba_{0.5}Al_{0.5}O_x$ due to the presence of more amorphous $Al_2O_3$ in the catalysts. Amorphous $Al_2O_3$ with abundant negative charges also favors the deposition and dispersion of Pt nanoparticles via a strong electrostatic interaction [39]. However, the mole ratio of Al increased to 3. As a result, the MOR activity of the $Ba_{0.33}AlO_x$ catalyst was lower than that of other catalysts, possibly because excessive Al contacts produce oxygen vacancy defects and cause the long-distance elongation of O–O bond [40]; thus, the MOR activity decreased. The SEM and TEM images of the $20\%Pt-Ba_{0.5}AlO_x/C$ catalyst are illustrated in Figures 7 and 8a. In Figure 8b, the Pt 4f XPS spectra of $20\%Pt/C$ and $20\%Pt-Ba_{0.5}AlO_x/C$ catalysts were assigned to the metallic state of $Pt^0$ and the oxide states of $Pt^{2+}$ and $Pt^{4+}$. The binding energy of the Pt 4f peaks of $20\%Pt-Ba_{0.5}AlO_x/C$ shifted positively compared with that of $20\%Pt/C$, and this shift could be attributed to the interactions between Pt and metal oxides [14] or metal-support interactions [41], indicating electron transfer [42] between Pt and $Ba_{0.5}AlO_x$ or atomically dispersed $Pt^0$ and Pt aggregates on metal oxide [43]. Moreover, the $Pt^0$ content of $20\%Pt-Ba_{0.5}AlO_x/C$ catalysts increased to 70% compared with that of 52.6% of $20\%Pt/C$ catalysts, thereby providing more available active Pt sites for methanol adsorption [14].

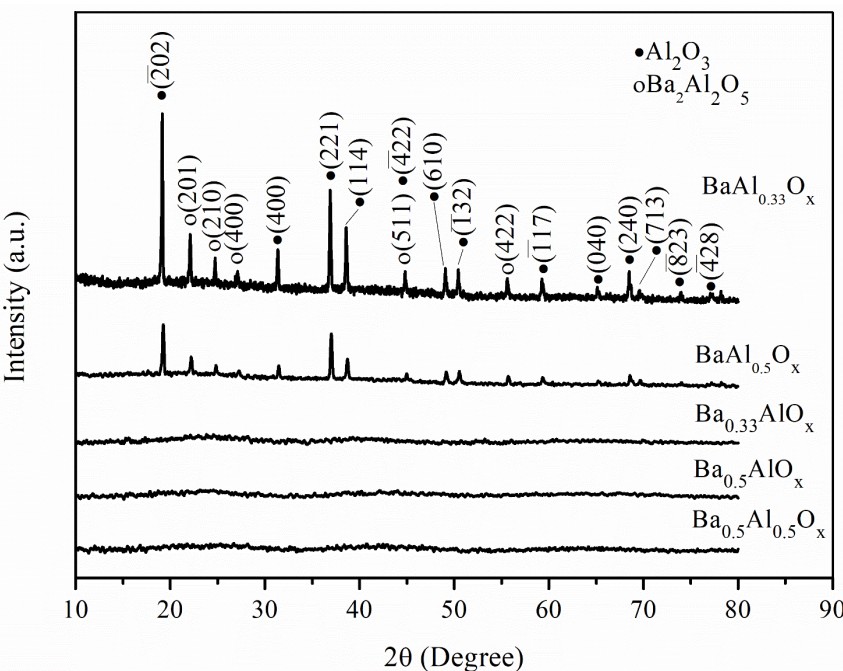

**Figure 6.** XRD of various $BaAlO_x$ powders prepared at different ratios of Ba and Al contacts.

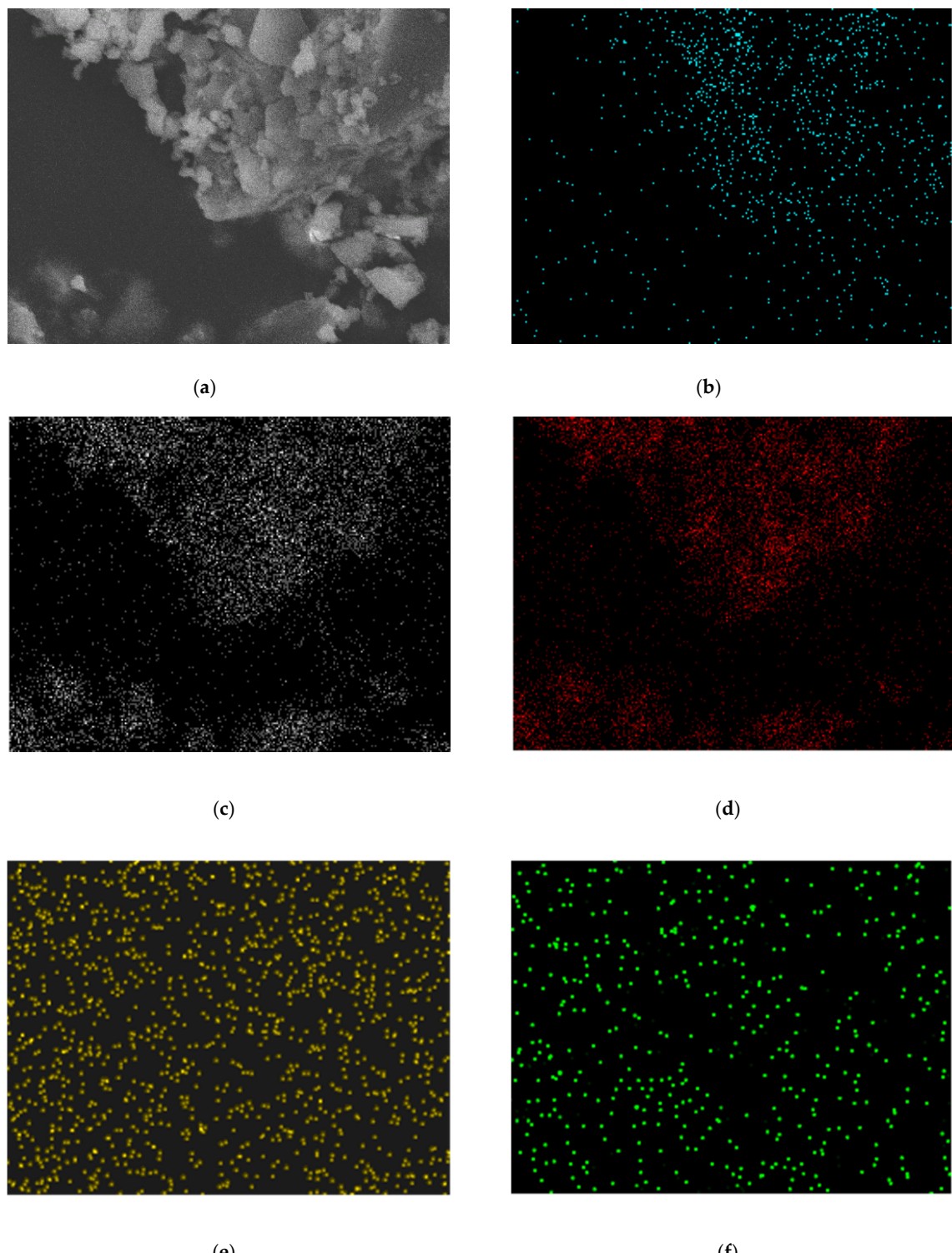

(a)

(b)

(c)

(d)

(e)

(f)

**Figure 7.** SEM mapping image of (**a**) 20%Pt-Ba$_{0.5}$AlO$_x$/C catalyst, (**b**) Ba, (**c**) Al, (**d**) O, (**e**) C, and (**f**) Pt element.

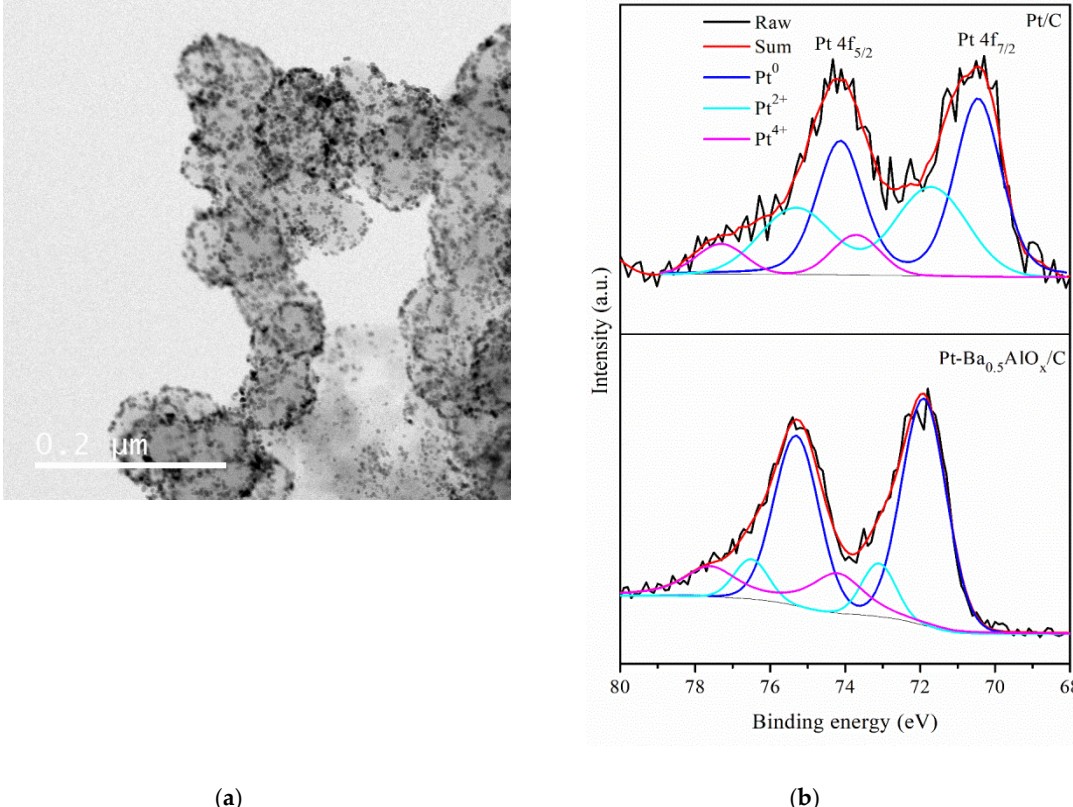

(**a**)　　　　　　　　　　　　　　　　　　　　(**b**)

**Figure 8.** (**a**) TEM image of 20%Pt-Ba$_{0.5}$AlO$_x$/C and (**b**) Pt 4f XPS of two catalysts.

The bifunctional mechanism of Pt-metal oxide/C catalysts depends on the synergistic contribution of bimetallic oxide on Pt catalysts to MOR [12] occurred by the OH adsorbed on the surface of the metal oxide that may oxidize the CO present on the surface of Pt [11,37,44]. Therefore, the bifunctional mechanism of MOR with 20%Pt-Ba$_{0.5}$AlO$_x$/C catalysts can be summarized in (1)–(4). In (1), methanol is initially adsorbed on Pt, and its methanolic proton is simultaneously lost to a basic oxide ion. In (2), the presence of Ba$_{0.5}$AlO$_x$ groups may induce OH$_{ads}$ species to react with CO$_{ads}$ on the Pt surface. As a result, CO$_2$ is produced, as shown in (3). Consequently, more active sites on the Pt surface are released, thereby facilitating methanol oxidation. A strongly bound intermediate is expected to be removed from the electrocatalyst surface by reacting with Ba$_{0.5}$AlO$_x$. In (4), Pt loaded on Ba$_{0.5}$AlO$_x$ may react with CO to form CO$_2$.

$$CH_3OH_{ad} \rightarrow CO_{ad} + 4H^+ + 4e^- \tag{1}$$

$$Ba_{0.5}AlO_x + H_2O \rightarrow Ba_{0.5}AlO_x\text{-}OH_{ads} + H^+ + e^- \tag{2}$$

$$Pt\text{-}CO_{ads} + Ba_{0.5}AlO_x\text{-}OH_{ads} \rightarrow Pt + Ba_{0.5}AlO_x + CO_2 + H^+ + e^- \tag{3}$$

$$Pt\text{-}Ba_{0.5}AlO_x + yCO \rightarrow Pt\text{-}Ba_{0.5}AlO_{x\text{-}y} + yCO_2 \tag{4}$$

The long-term electrochemical stability of 20%Pt/C and 20%Pt-Ba$_{0.5}$AlO$_x$/C catalysts in MOR was investigated by performing 1000 continuous potential cycles in 1 M CH$_3$OH + 0.5 M H$_2$SO$_4$ solution (Figure 9). Clearly, the current density of both samples decayed when the number of cycles increased. The maximum current density of the 20%Pt-Ba$_{0.5}$AlO$_x$/C catalyst was still higher than that of the 20%Pt/C catalyst after 1000 cycles. Therefore, the long-term electrochemical stability of 20%Pt-Ba$_{0.5}$AlO$_x$/C catalysts was better than that of 20%Pt/C catalysts, which was attributed to the efficient removal of the intermediate species benefited from the synergetic effect [19] between Pt and Ba$_{0.5}$AlO$_x$.

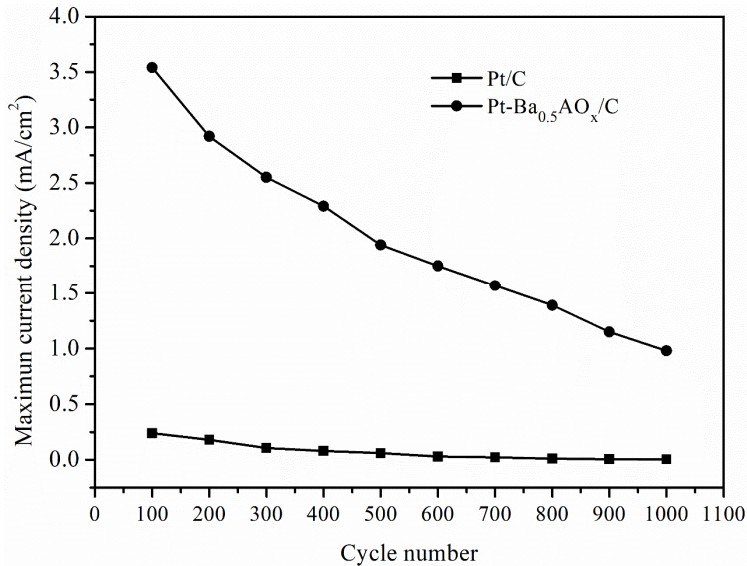

**Figure 9.** The stability test of 20%Pt/C and 20%Pt-Ba$_{0.5}$AO$_x$/C catalysts.

## 3. Materials and Methods

### 3.1. Chemicals and Materials

The following chemicals were used in this study: aluminum nitrate nonahydrate (Al(NO$_3$)$_3$·9H$_2$O), barium nitrate (Ba(NO$_3$)$_2$), and sodium hydroxide (NaOH; Showa Chemical Industry Co., Ltd., Tokyo, Japan); chloroplatinic acid (IV) hexahydrate (H$_2$PtCl$_6$·6H$_2$O; Alfa Aesar, Lancashir, UK); ethylene glycol (C$_2$H$_6$O$_2$; J.T. Baker®, Phillipsburg, OH, USA); sulfuric acid (H$_2$SO$_4$) and methanol (CH$_3$OH; Aencore Chemical Co., Ltd., Surrey Hills, Australia); ethanol (C$_2$H$_5$OH, 95%; ECHO Chemical Co., Ltd., Miaoli, Taiwan); Vulcan XC-72 carbon black (BET surface area of 250 m$^2$/g; Cabot Co., Alpharetta, GA, USA); 20 wt.% Nafion solution (DuPont Co. Ltd., Wilmington, DE, USA).

### 3.2. Preparation of Pt–Barium Aluminum Oxide/C Catalysts

Amorphous BaAlO$_x$ catalysts were synthesized through a simple polyol thermal method. In a typical synthesis, 100 mL of 0.1 M Al(NO$_3$)$_3$·9H$_2$O and 100 mL of 0.1 M Ba(NO$_3$)$_2$ were mixed with 40 mL of ethylene glycol under magnetic stirring and heated at 120 °C for 1 h, thereby producing a white precipitate. The product was cooled and centrifuged at 5000 rpm three times for 15 min each. After centrifugation, the solution was poured, and the precipitate was washed with distilled water, placed in a dish, and heated in an oven at 80 °C for 24 h. The 20 wt.% Pt-BaAlO$_x$/C prepared by 0.04 g of amorphous BaAlO$_x$ powder was mixed with 0.04 g of Vulcan XC-72 carbon black (1:1 weight ratio), dispersed in 40 mL of ethylene glycol in a 100 mL beaker, and ultrasonicated for 15 min under magnetic stirring for 1 h to form a uniform suspension. Afterward, 10 mL of 5.2 mg/mL H$_2$PtCl$_6$ in ethylene glycol solution was supplemented to the suspension. After the mixture was stirred vigorously for 1 h, the pH of the suspension was adjusted to 11 by adding 0.5 M NaOH to ethylene glycol dropwise. The mixture was heated to 140 °C, and a reflux system was set up under reflux for 3 h. After the mixture was cooled, the precipitates were centrifuged at 5000 rpm three times for 15 min each. The solution was poured, and the precipitate was washed with distilled water, placed in a dish, and dried in an oven at 80 °C for 24 h. Finally, 13.6 wt.% ± 1.2 wt.%Pt metal content in 20 wt.%Pt-BaAlO$_x$/C and 20 wt.%Pt/C catalysts was confirmed by an inductively-coupled plasma optical emission spectrometer (ICP-OED, Agilent 725, Santa Clara, CA, USA).

### 3.3. Material Characterization

The morphological characteristics of the powder were analyzed through scanning electron microscopy (SEM) by using a JED 2300 instrument (JEOL Ltd., Peabody, MA, USA). The crystallographic patterns of the powder were recorded through X-ray diffraction (XRD) in a Rigaku (Tokyo, Japan) ultima IV rotating anode diffractometer with a Ni-filtered Cu–K radiation source (wavelength of 1.54 Å). X-ray photoelectron spectroscopy (XPS) was performed with VG Scientific ESCALAB 250 (Thermo Fisher Scientific Inc., Loughborough, UK) equipped with a dual Al X-ray source operated at 200 W and 15 kV, 650 μm of beam size, and a hemispherical analyzer operating in a constant analyzer energy (CAE) mode. The base pressure in the analyzing chamber was maintained at $10-10$ mbar. Data profiles were subjected to a nonlinear least-squares curve-fitting program with a Gaussian–Lorentzian production function and processed using the Casa XPS program (Casa Software Ltd., Teignmouth, UK). An adventitious C1s binding energy of 284.9 eV was set as the reference binding energy for charge correction. Transmission electron microscopy (TEM) was carried out with a JEM-2100 electron microscope with an acceleration voltage of 200 kV (JEOL Ltd., Peabody, MA, USA). The surface areas of the powder were measured with the conventional Brunauer–Emmett–Teller (BET) method (ASAP 2020 model; Micromeritics, Norcross, GA, USA).

### 3.4. Electrochemical Measurements

Electrochemical measurements were carried out on a computer-controlled CHI 608E electrochemical analysis instrument (CH Instruments, Inc., Bee Cave, TX, USA) in a three-electrode electrochemical cell at room temperature. A Pt spiral and an Ag/AgCl electrode were used as a counter electrode and a reference electrode, respectively. A working electrode was prepared with 20 mg of catalyst dispersed in 1.8 mL of ethanol and mixed with 0.2 mL of 5 wt.% Nafion solution via sonication for 30 min. Then, 1 μL of catalyst suspension was spread onto the surface of the glassy carbon (GC) electrode with a diameter of 3 mm and dried. Cyclic voltammetry (CV) was utilized to investigate methanol oxidation within a potential range of −0.2 V to 0.9 V at 50 mV/s in 0.5 M $H_2SO_4$ containing 1 M $CH_3OH$. Hydrogen adsorption-desorption peaks were obtained in 0.5 M $H_2SO_4$, which was deaerated with ultrapure $N_2$ gas for 30 min before each experiment. The long-term durability of the catalysts was tested by conducting 1000 continuous potential cycles between 0.05 and 1.20 V at 50 mV/s. Electrochemical impedance spectra (EIS) were obtained at 0.4 V and 100 kHz–0.01 Hz in 1 M $CH_3OH$ and 0.5 M $H_2SO_4$ mixture. The electrochemically active surface area (ECSA) could be obtained using the following Equation: ECSA $(m^2/g) = Q_H/(0.21$ mC/$m^2 \times$ mPt) [45]. $Q_H$ (mC/$cm^2$) is the average of integrating the hydrogen adsorption and desorption area (mA/$cm^2$·V) by utilizing Origin 8.0 (OriginLab Corporation, Northampton, MA, USA) and dividing the screen rate (V/sec). Furthermore, 0.21 mC/$cm^2$ was set for the oxidation of a hydrogen monolayer on a Pt electrode [44], and corresponding to 20%Pt loaded on the BaAlO$_x$ catalysts on a disk electrode, it was 0.24 g/$m^2$.

### 4. Conclusions

Novel BaAlO$_x$ powder was synthesized through a simple polyol thermal method to prepare 20%Pt-BaAlO$_x$/C catalysts for enhancing MOR in DMFC. The uncalcined amorphous $Ba_{0.5}AlO_x$ catalysts exhibited a dramatically enhanced MOR activity, depending on their large surface area and abundant oxygen vacancies. The synergistic effects of amorphous $Ba_{0.5}AlO_x$ on Pt catalysts induced the bifunctional mechanism of 20%Pt-$Ba_{0.5}AlO_x$/C catalysts for MOR. As a result, Pt-$Ba_{0.5}AlO_x$/C catalysts possessed the best MOR performance. The 20%Pt-$Ba_{0.5}AlO_x$/C catalysts demonstrated the largest maximum current density of 3.89 mA/$cm^2$, the largest ECSA of 49.83 $m^2$/g, and the smallest $R_{ct}$. These catalysts also had better long-term durability and stability than 20%Pt/C catalysts after 1000 cycles. Therefore, 20%Pt-$Ba_{0.5}AlO_x$/C catalysts were optimum anode electrode candidates for DMFCs.

**Author Contributions:** Writing—original draft preparation, writing—review and editing, supervision, and project administration, T.H.C.; formal analysis and data curation, W.-Y.H.; validation, J.-W.H. and Y.-S.C. All authors have read and agreed to the published version of the manuscript.

**Funding:** This research was funded by the Ministry of Science and Technology, grant number MST 105-2221-E-239-010.

**Conflicts of Interest:** The authors declare no conflict of interest.

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
