# Peer review of "Pt-Amorphous Barium Aluminum Oxide/Carbon Catalysts for an Enhanced Methanol Electrooxidation Reaction"

_catalysts, doi:10.3390/catal10060708_

Round 1
Reviewer 1 Report
Manuscript ID: catalysts-833328
Type of manuscript: Article
Title: Pt-amorphous barium aluminum oxide/Carbon catalysts for an enhanced
methanol electrooxidation reaction
Summary:
Amorphous barium aluminum oxide was synthesized, mixed with carbon black and then used as support of Pt catalyst, in order to enhance the activity of a methanol electrooxidation reaction. BaAlOx was exposed to different calcining temperatures (uncalcined, 200 °C and 400 °C), finding that the maximum current density and ECSA for MOR were higher in the case of the uncalcined catalyst. XRD and TEM characterization showed amorphous structures in all cases, but different particle size distribution and different surface area. In particular, calcination increased the particle size and decreased the surface area, explaining why the activity was higher in the case of the uncalcined sample. XPS analyses showed similar BE for O and Ba, while the binding energy of Al showed a positive shift when samples were calcined. Authors investigated also the effect of the Ba and Al precursors ratio, finding that the presence of Ba in BaAlOx facilitated the oxidation of chemisorbed species enhancing the activity. The Ba/Al ratio influenced also the crystallinity: the activity of the crystalline BaAlOx was lower than the amorphous one, which has more oxygen vacancies. Moreover, the shift in the BE of Pt could be attributed to the interaction between Pt and metal oxides. Finally, the long-term electrochemical stability of the Pt-BaAlOx/C was studied showing that it was higher than that of Pt/C.
Major revisions
Comments for the authors:
The research is complete and rather well presented. However, there are some points which need to be clarified or improved.
Abstract:
- I would avoid the use of terms such as “carefully” and “significantly”.
- Change “significantly enhanced compared with” with more quantitative data, in order to give an idea of how is the enhancement.
- “This reaction….reaction resistance” – I would remove this sentence from the abstract.
- 20% Pt: this is the only case in which the % of Pt is reported. 20 % wt.? considering the Pt precursor or Pt as single species? Moreover, is 20 % loading considered a high loading amount of Pt or not? Discuss this.
Introduction:
- The text refers to the “high loading amount of Pt to maintain the DMFC performance”. The 20 % Pt used in this research should be discussed.
- Line 41: Al2O3 is usually/often modified by…
- The aim of this research should be emphasized, also discussing why enhance MOR is important.
Results and discussion:
- Use the same significant digits or, better, do not report 0.657, 0.65, 0.649 but only a single mean value.
- Add a reference after the sentence at line 62: “..possesses high electrocatalytic activity” (ref).
- Explain better the link between the particle size and the activity in MOR.
- Surface area values are reported with too much digits. In my opinion you can also avoid to report the error, and use mean values.
- Lines 93-96: the sentence is not clear, please reformulate.
- Why there is a positive shift of the BE of Al after calcination? What is it due to? Can it be related to the shift of the BE of Pt? this point must be discussed in more detail.
- It seems that from the paragraph 2.2 the font size is different.
- Line 133: which – to correct.
- Line 169-172: check the English and reformulate the sentence.
- The improvement of the electrocatalytic activity of MOR due to the presence of oxygen vacancies should be discussed. How can you prove that the oxygen vacancies of the Ba0.5AlOx catalyst are more than those of the Ba0.5Al0.5Ox?
- In order to prove that there is an interaction between Pt and metal oxide, BE shift must be discussed in more detail, proving that there could be an electronic exchange between the two species.
- Figure 9: the authors compared the Pt/C with the Pt-Ba0.5AlOx/C. Which are the differences between the two samples in terms of: particle size, exposure, Pt loading, crystallinity…. The Pt/C used as reference must be characterized to be compared.
Author Response
1. I would avoid the use of terms such as “carefully” and “significantly”. Change “significantly enhanced compared with” with more quantitative data, in order to give an idea of how is the enhancement.
Replay:
These words as “carefully” and “significantly” have been deleted, and give quantitative data at line 19 to 20.
2. “This reaction….reaction resistance” – I would remove this sentence from the abstract.
Replay:
We have been removed this sentence.
3. 20% Pt: this is the only case in which the % of Pt is reported. 20 % wt.? considering the Pt precursor or Pt as single species? Moreover, is 20 % loading considered a high loading amount of Pt or not? Discuss this Introduction:
Replay:
20%Pt-BaAlOx/C is represent 20 wt.% Pt metal content loading onBaAlOx/C. However, the processing of Pt loading on BaAlOx/C preparation would be loss some Pt precursor. For this study, 13.6 wt.% ± 1.2 wt.% Pt metal content in 20 wt.%Pt-BaAlOx/C and 20 wt.%Pt/C catalysts confirmed using by inductively coupled plasma optical emission spectrometer, which has been added at line 262 to 263 of section 3.2.
In addition, a commercial 20%Pt/C catalyst is most commonly used for as cathode catalysts of polymer electrolyte fuel cell (PEMFC), DMFC and other fuel cells. Therefore, 20%Pt loading was be a reference material in this study. This illustrate has been added at line 33 to 34 of Introduction.
4. The text refers to the “high loading amount of Pt to maintain the DMFC performance”. The 20 % Pt used in this research should be discussed.
Replay:
“20%Pt” has add in all of catalysts in the text.
5. Line 41: Al2O3 is usually/often modified by…
The aim of this research should be emphasized, also discussing why enhance MOR is important.
Replay: “Al2O3 is usually/often modified by…” has been corrected, and enhance MOR discursion has been added at line 43 to 45.
Results and discussion:
6. Use the same significant digits or, better, do not report 0.657, 0.65, 0.649 but only a single mean value.
Reply: We have correct these values.
7. Add a reference after the sentence at line 62: “..possesses high electrocatalytic activity” (ref).
Reply:
We have add a reference after the sentence at line 65.
8. Explain better the link between the particle size and the activity in MOR.
Reply:
Please see the line 97 to 100
9. Surface area values are reported with too much digits. In my opinion you can also avoid to report the error, and use mean values.
Reply:
We have correct this data of surface area values.
10. Lines 93-96: the sentence is not clear, please reformulate.
Reply:
Please see the line 101 to 103
11. Why there is a positive shift of the BE of Al after calcination? What is it due to? Can it be related to the shift of the BE of Pt? this point must be discussed in more detail.
Rerly:
The Al XPS of Fig. 4 (c) obtained without Pt loading on BaAlOx, the positive shift of the BE of Al after calcination due to Al(OH)3 or AlO(OH) formation. The result is similar to Reddy et al. report.
Reddy et al., Ceramics International, 40, (2014), 11099–11107.
12. It seems that from the paragraph 2.2 the font size is different.
Reply: We have been modified this mistake.
13. Line 133: which – to correct.
Reply:
We have been modified this mistake.
14. Line 169-172: check the English and reformulate the sentence.
Reply:
These sentence have been reformulated at line 171-175.
15. The improvement of the electrocatalytic activity of MOR due to the presence of oxygen vacancies should be discussed. How can you prove that the oxygen vacancies of the Ba0.5AlOx catalyst are more than those of the Ba0.5Al0.5Ox?
Reply:
About the improvement of the electrocatalytic activity of MOR due to the presence of oxygen vacancies have been explained at line 179-181.
In additioin, we have a difficult to confirm that the oxygen vacancies of the Ba0.5AlOx catalyst are more than those of the Ba0.5Al0.5Ox through some experiments. However, Pask et al. reported that the concentration of oxygen vancancies increased by Al dopant concentration in TiO2 increased. Therefore, this study conjecture high Al contect of Ba0.5AlOx catalyst have more oxygen vacancies than Ba0.5Al0.5Ox.
Joseph A. Pask, Anthony G. Evans Eds, Ceramic Microstructures '86: Role of Interfaces, 1986, page 529
16. In order to prove that there is an interaction between Pt and metal oxide, BE shift must be discussed in more detail, proving that there could be an electronic exchange between the two species.
Reply:
The reason of interaction between Pt and metal oxide has been showed at line193-195
17. Figure 9: the authors compared the Pt/C with the Pt-Ba0.5AlOx/C. Which are the differences between the two samples in terms of: particle size, exposure, Pt loading, crystallinity…. The Pt/C used as reference must be characterized to be compared.
Reply:
When we compared the long-term electrochemical stability of the Pt/C with the Pt-Ba0.5AlOx/C, which is based on same loading and same preparation processing of Pt on carbon and Ba0.5AlOx/C mixture. The long-term electrochemical stability of 20%Pt-Ba0.5AlOx/C catalysts was better than that of 20%Pt/C catalysts, which is attributed to the efficient removal of the intermediate species benefited from the synergetic effect between Pt and Ba0.5AlOx. Therefore, Ba0.5AlOx is an important factor for the catalytic performance.
Reviewer 2 Report
The article entitled "Pt-amorphous barium aluminum oxide/Carbon catalysts for an enhanced methanol electrooxidation reaction" is well written and worth to be accepted by Catalysts.
I would suggest some minor/editorial changes:
- adding descri Eq.1 - Eq.4 to equations; now the description (1) - (4) is confusing
- captions of figures should be written with smaller type
Author Response
(1) adding descri Eq.1 - Eq.4 to equations; now the description (1) - (4) is confusing
Reply:
We have been deleted these “Eq.” at line 221-224.
(2) captions of figures should be written with smaller type
Reply:
Captions of Figs.1, 5, and 9 have been modified with short.
Round 2
Reviewer 1 Report
Dear Author,
after this revision, the paper is improved.
However, there are still some parts not clear enough or not well presented.
my observations below:
Check the sentence:
“Due to more oxygen vacancies are generated lead to enhance the surface oxygen concentration for metal oxide catalysts that provide more metal oxide OHads bond [37], which boosts the conversion of COads to CO2 on the Pt- metal oxide/C catalyst [38].”
It should be reformulated as: “the more oxygen vacancies enhance the surface oxygen concentration for metal oxide catalysts that provide more metal oxide OHads bond. For this reason, the conversion of COads to CO2 is higher on the Pt-metal oxide/C catalyst.”
I think is better in this form.
About the XPS analyses, when you write:
“indicating electron transfer [42] between Pt and Ba0.5AlOx or atomically dispersed Pt0 195 and Pt aggregates on metal oxide [43]”
I suggest to add a table including the Binding Energy values calculated from the XPS deconvolution.
About the synergetic effect:
“which is attributed to the efficient removal of the intermediate species benefited from the synergetic effect [19] between Pt and Ba0.5AlOx.”
As a proof of the synergetic effect, is the monometallic catalyst less active then the bimetallic ones, considering the same mol of metal?
Could you explain in more detail why you can say that “there is a synergetic effect”?
Author Response
Please refer to the attached "Response tp Revoewer 1 Comments" for more details
